# The Potential Mediation of the Effects of Physical Activity on Cognitive Function by the Gut Microbiome

**DOI:** 10.3390/geriatrics5040063

**Published:** 2020-09-25

**Authors:** Victoria Sanborn, John Gunstad

**Affiliations:** 1Department of Psychological Sciences, Kent State University, Kent, OH 44240, USA; jgunstad@kent.edu; 2Brain Health Research Institute, Kent State University, Kent, OH 44240, USA

**Keywords:** physical activity, gastrointestinal microbiome, cognition, dementia

## Abstract

The population of older adults is growing dramatically worldwide. As older adults are at greater risk of developing disorders associated with cognitive dysfunction (i.e., dementia), healthcare costs are expected to double by 2040. Evidence suggests dementia may be slowed or prevented by lifestyle interventions, including physical activity (PA). PA is associated with improved cognitive function and may reduce risk for dementia by mitigating known risk factors (i.e., cardiovascular diseases) and/or by enhancing neurochemical processes. An emerging area of research suggests the gut microbiome may have similar neuroprotective effects. Altering the gut microbiome has been found to target physiological processes associated with dementia risk, and it influences gut-brain-microbiome axis signaling, impacting cognitive functioning. The gut microbiome can be altered by several means (i.e., disease, diet, prebiotics, probiotics), including PA. As PA and the gut microbiome independently influence cognitive function and PA changes the composition of the gut microbiome, cognitive improvement due to PA may be partially mediated by the gut microbiome. The present article provides an overview of the literature regarding the complex associations among PA, cognitive function, and the gut microbiome, as well as their underlying biological mechanisms. A comprehensive, theoretical model integrating evidence for the potential mediation is proposed.

## 1. An Aging Population with Risk of Cognitive Decline

The population of older adults is growing worldwide. Predictions for the United States (US) suggest that the population of older adults will increase to 55 million by 2020 and to 70 million by 2030 [1]. Worldwide, there are currently 617 million older adults, a population projected to increase to 1.6 billion by 2050 [2]. In addition to being at increased risk for chronic illness [3], older adults experience cognitive aging and are at greater risk of developing disorders associated with cognitive dysfunction, such as dementia.

Some decline in mental abilities is normal with advancing age, including difficulties with fluid tasks (e.g., complex attention, memory recall) [4]. However, preservation of crystalized abilities (e.g., recognition memory, language) allows most adults to maintain high levels of daily functioning until at least their mid-60s [4,5]. Age-related changes in cognitive functioning are attributed to normal changes in brain structure and function, including reductions in global brain volume [6,7], reductions in the frontal and parietal lobes [8], and atrophy in the medial and hippocampal regions [9]. Whereas cognitive aging is inevitable, disorders associated with cognitive dysfunction are pathological and may be prevented.

Older adults are also at increased risk of developing disorders associated with cognitive dysfunction (i.e., dementia) [10]. Dementia involves a decline in cognitive functioning and/or neuropsychiatric symptoms resulting in the loss of functional abilities [11]. Preceding dementia, many persons may be identified as having mild cognitive impairment (MCI), which involves objective evidence of cognitive impairment beyond normal cognitive aging with preservation of daily functioning [12]. It is estimated that 16–20% of persons aged 60 to 90 meet criteria for MCI [13] and have an increased likelihood of developing dementia.

The two most common types of dementia are Alzheimer’s disease and vascular dementia. Alzheimer’s disease (AD) is the most common type with recent estimates suggesting prevalence of 5.4 million persons in the US alone [14] and 35.6 million persons globally [15]. AD is characterized by memory problems and other cognitive impairments associated with an accumulation of β-amyloid (Aβ) plaques and neurofibrillary tangles in the brain [16]. Vascular dementia (VaD) is the second most common type of dementia with an estimated 5.94 × 10^5^ cases in the US [17]. VaD and AD are thought to have overlapping pathologies relating to cerebrovascular risk factors [18]. Whereas AD typically involves memory impairments along with other cognitive changes, VaD is characterized by more variable presentation of cognitive impairments dependent on the affected areas of the brain, and it is characterized by either a rapid onset of impairments secondary to stroke or a stepwise progression of minor infarcts. However, VaD typically impacts frontostriatal circuits resulting in reductions in attention and executive functions [19], as well as neuropsychiatric features including depression and apathy [20].

The current prevalence of all-cause dementia is alarmingly high. Recent estimates indicate that 46.8 million persons have dementia worldwide, a number that is projected to increase to 74.7 million by the year 2030 [21]. As prevalence of dementia is expected to increase with the world’s aging population, these changes will have a notable impact on individual and public healthcare costs.

### The Costs of Cognitive Decline

An increasing population of older adults at greater risk for dementia has notable consequences. At the individual level, quality of life (QoL) declines with impaired cognitive abilities. QoL refers to psychological, social, and physical domains of health impacted by a person’s perceptions, beliefs, experiences, and expectations [22]. QoL can be measured using objective and subjective assessment [22]. Compared to persons without cognitive impairment, persons with MCI, those with AD, and/or those living in residential care homes report worse physical and emotional health, greater daily limitations, and less independence in daily functioning [23,24,25].

In addition to the personal costs incurred by neurodegenerative disorders and cognitive dysfunction, public healthcare costs will increase as well. Currently, estimates suggest that healthcare costs due to dementia alone total $157 to $215 billion in the US [26], projected to double by 2040 [27]. Globally, recent consensus data estimate that costs of dementia currently total $818 billion, with the greatest costs in high-income countries [28].

Given the increasing population of older adults and projected surges in related costs, the World Health Organization (WHO) has recognized dementia as a public health priority [29]. As dementia is not reversible [30] and clinical trials seeking to treat symptoms of dementia have shown little progress [31], a recent call for preventive measures has arisen [32]. Although there is no cure for dementia, some lifestyle interventions have been shown to mitigate its effects, including increasing physical activity.

## 2. Physical Activity

### 2.1. Defining Physical Activity

The term physical activity (PA) can refer to reducing sedentary behavior, increasing light physical activity, and purposeful exercise. To promote clarity for the current review, PA is defined using metabolic equivalent units (METs). METs refer to the amount of oxygen consumed during PA; 1 MET refers to the amount of oxygen consumed at rest, typically 3.5 mL O_2_/kg/min [33]. MET is a simple, commonly used standard of measurement in North America that can be used to quantify energy expenditure and functional capacity, and identify safe, physical activities for individuals [33].

Sedentary behavior (SB) includes activities that do not increase energy expenditure significantly beyond resting level (e.g., sitting, sleeping, lying down, watching television) [34]. Such behaviors are operationally defined as involving energy expenditure of 1 to 1.5 METs [34]. A greater amount of time spent engaged in SB, controlling for exercise, has been independently associated with a variety of poor health outcomes [35] including all-cause morbidity and mortality from cardiovascular diseases [36], as well as obesity and metabolic syndrome [37].

Light PA refers to behaviors involving any movement (e.g., writing, cooking, washing dishes, slow walking) [38] and is defined as 1.6 to 2.9 METs [34]. Even when accounting for other key demographic and medical risk factors, a greater amount of time spent in light PA has been associated with reduced likelihood of all-cause mortality up to 10 years later [39]. More time spent in light PA has also been associated with specific health factors including reduced fasting glucose levels [40], less arterial stiffness [41], and lower body mass index (BMI), waist circumference, insulin resistance, and c-reactive protein (CRP) [42].

Unlike SB and light PA, exercise is intentional and planned [38]. Exercise can range in intensity from light PA (as described above) to moderate through vigorous PA. Moderate PA involves energy expenditure of 3 to 6 METs while vigorous PA involves energy expenditure of 6+ METs [33]. Moderate exercise includes activities such as weightlifting, yoga, and bicycling, and vigorous exercise includes activities such as running, swimming, and tennis [33]. Moderate-to-vigorous physical activity (MVPA) has been found to reduce all-cause morbidity in older adults by 22% [43].

In addition to the health benefits obtained by decreasing SB and/or increasing light PA and exercise, PA has also been found to improve cognitive functioning. The potential mechanisms for this association are reviewed later in this text.

### 2.2. Physical Activity and Cognitive Function

PA, or lack thereof, appears to significantly impact cognitive functioning in older adults. Greater amount of time spent in SB has notable, deleterious effects on middle-aged and older adults’ cognitive functioning [44,45,46]. Some interventions attempted to reduce SB using a variety of techniques (e.g., goal setting, accelerometer counts, increasing standing time) [47,48,49] which show some promise, although they largely examined feasibility and have yet to show notable results for impact on cognitive functioning [50].

Light PA appears to have more notable effects on cognitive function, although results are mixed. Some studies showed positive effects for a greater amount of time spent in light PA for cognitive functioning in middle-aged to older adults, including better global cognition and neuropsychological performance [51,52,53]. However, some studies did not corroborate these results, finding that middle-aged and older adults who engaged in more light PA did not show improved cognitive function over time [54,55]. Intervention studies mirrored these mixed findings, with some showing that increasing light PA is associated with greater improvements in cognitive functioning than inactivity [56], while others failed to find this association [57].

MVPA appears to have a more significant impact on cognitive function in older adults than SB or light PA. Self-reported MVPA has been associated with lower risk for cognitive impairment, better executive functioning [54], improved cognitive function [58], and reduced risk for cognitive decline [59]. MVPA interventions showed similar associations. Increasing the amount of exercise was found to improve cognitive functioning in older adult women [60], community-dwelling older adults [61], and persons with diagnosis of MCI or dementia [62]. Systematic reviews of randomized control trials (RCTs) involving exercise confirm this pattern, showing notable benefits of MVPA on global cognitive functioning [63,64] with the greatest gains identified for executive functioning [65].

There is much evidence to suggest that greater intensity of PA and increasing amount of time spent in PA are beneficial for cognitive function in older adults. Although research examining the effects of reducing SB or increasing light PA is mixed, the association between more MVPA and better cognitive function has been repeatedly confirmed.

### 2.3. Potential Mechanisms for Effects of Physical Activity on Cognitive Function

There are several potential mechanisms that may explain the positive influence of PA on brain health and cognitive functioning in older adults, including indirect pathways involving mitigation of disease, as well as direct pathways involving promotion of neurochemical processes.

#### 2.3.1. Indirect Pathways between PA and Cognitive Function

PA improves diseases and related processes known to impact cognitive functioning, including cardiovascular function, diabetes, obesity, and depression. Two cardiovascular pathways through which PA impacts cognitive function are hypertension and poor vascular function.

PA can prevent hypertension. Prevalence of hypertension increases with age; approximately 65% of adults aged 65 to 74 and 76% of adults aged 75+ are diagnosed with hypertension [66]. Hypertension has been consistently associated with cognitive dysfunction, including poorer psychomotor speed, set-shifting [67], and word fluency [68], as well as increased likelihood of developing dementia [69,70]. However, research showed that hypertension can be prevented with increased engagement in both anaerobic and aerobic exercise [71,72,73,74,75].

Exercise also improves vascular function. Vascular function refers to the balance of relaxation and contraction of the endothelium [76,77]. Poor vascular function is often indicated by the presence of atherosclerosis [78] and development of vascular diseases including stroke and myocardial infarction [79]. Atherosclerosis [80] and arterial stiffness [81,82] have both been associated with cognitive dysfunction. Indicators of good vascular function, including low arterial stiffness, low plasma endothelin-1, and high plasma nitric oxide, were all shown to improve with physical exercise [77,83,84,85,86].

In addition to cardiovascular diseases, exercise has been found to improve factors associated with diabetes. Four types of diabetes have been identified, including type 1, type 2, gestational, and due to other causes, all of which lead to insulin resistance or total insulin deficiency [87]. Prevalence of diabetes in older adults is greater than 25% [88] and has been associated with MCI, dementia [89], cognitive decline [90], and poorer overall cognitive function [91,92,93]. In type 2 diabetes specifically, exercise was found to ameliorate metabolic dysfunction by reducing hemoglobin A1c (HbA1c) levels in older adults [94,95].

Exercise also improves obesity-related factors. In the US, approximately one-third of older adults meet criteria for having obesity [96], with higher rates identified in Black and Hispanic persons [97], as well as persons of lower socioeconomic status (SES) [98]. Obesity has been associated with cognitive dysfunction [99], increased risk for neurodegenerative disorders [100], and reduced brain volume [101,102]. However, there is some evidence for the “obesity paradox”, which suggests that obesity during middle age is worse for cognitive outcomes than obesity in older adulthood [103,104]. Exercise was found to improve obesity in older adults, evidenced by weight loss [105], smaller waist circumference [106], lower waist-to-hip ratio, and lower BMI [107].

Exercise has also been found to improve psychological disorders associated with cognitive dysfunction, including depression. One to five percent of community-dwelling older adults and 13.5% of older adults requiring home care are thought to meet criteria for depression [14]. Depression has been associated with poorer executive functioning, processing speed, and memory in older adults [108], as well as accelerated rates of cognitive decline [109]; some research suggests depression may be a risk factor for development of dementia [110]. Exercise was found to improve depressive symptoms [111,112] with the greatest benefits identified using moderate-intensity aerobic exercise [113].

#### 2.3.2. Direct Pathways between PA and Cognitive Function

PA has also been shown to improve neurochemical processes and neuroanatomical structure. One such neurochemical is brain-derived neurotrophic factor (BDNF). BDNF is a neurotrophin [114] that helps regulate synaptic transmission and plasticity [115]. Reduced levels of BDNF are identified in the normal aging process [116], MCI [117], and AD [118,119], and have been associated with poorer verbal memory, information processing speed [117], and global cognitive function [120]. In mice, exercise was found to increase proteins that regulate BDNF [121]. In older adults, exercise was found to promote BDNF [122], whereas inactivity reduces it [123]. These results extend to persons with depression who show greater BDNF increases from exercise than cognitive training or mindfulness [124].

In additional to neurochemicals, exercise promotes neuroanatomical structural changes including neuroplasticity, neurogenesis, and angiogenesis. Neuroplasticity refers to the ability of neurons to reorganize, change [125], and signal efficiently [126]. Exercise promotes neuroplasticity by improving long-term potentiation (LTP) in the dentate gyrus of the hippocampus (known for its role in learning and memory) due to its effects on dendritic length, spine density, dendritic complexity, and neural progenitor cell proliferation (i.e., neural cells that divide a limited number of times) [127]. BDNF appears to modulate the effect of exercise on neuroplasticity by facilitating LTP and related signal transduction pathways [128].

Exercise has also been found to promote neurogenesis. Neurogenesis refers to the addition of new neurons to previously existing neuronal circuits [125]. Through this process, exercise was shown to increase hippocampal volume and improve cognitive function [116,129]. BDNF [130], insulin-like growth factor (IGF-1), and vascular endothelial growth factor (VEGF) are all active in neurogenesis [131]. During exercise, IGF-1 and VEGF are initially induced in the periphery and subsequently cross the blood–brain barrier (BBB) [131] to impact the brain. When IGF-1 and VEGF are blocked from the brain, the survival-promoting effect of exercise is reduced, indicating their effects are necessary to obtain the maximum neuronal benefits of exercise [132].

The effects of exercise on IGF-1 and VEGF promote angiogenesis as well. Angiogenesis refers to the growth of new blood vessels in the brain [133]. IGF-1 is necessary for exercise-induced angiogenesis in the brain [134], while increases in VEGF affect the proliferation, adhesion, survival, and capillary tube formation of vascular endothelial cells specifically [135]. Exercise improves markers of angiogenesis (i.e., density of micro vessels, cerebral blood flow) and increases levels of VEGF in older adult rats [136,137] and macaques [138]. Furthermore, in healthy older adults, higher levels of self-reported aerobic activity have been associated with a greater number of small-caliber vessels [139].

Exercise also reduces inflammation. Chronic inflammation is associated with neurodegeneration [140,141], cognitive decline [142], and dementia severity [143], as well as abnormalities on neuroimaging (e.g., reduced gray matter) [144]. Chronic inflammation also has deleterious effects on cognitive function in otherwise healthy adults [145]. The negative effects of chronic inflammation on the brain may be due to peripheral inflammatory markers (e.g., interleukin 6) crossing the BBB into the brain, which modulate central inflammatory responses detrimental to brain health [146]. Additionally, higher levels of proinflammatory cytokines were found to impair IGF-1 signal transduction and BDNF signaling in the brain, known to be important for maintaining cognitive functioning [131]. Exercise appears to counteract the effects of chronic inflammation in older adults by reducing key inflammatory markers [147,148,149].

There is much evidence to suggest that PA indirectly (i.e., counteracting disease) and directly (i.e., promoting neuroanatomical processes) improves brain outcomes. A new, less studied mechanism that may also impact the brain and cognitive function is the gut microbiome.

## 3. The Gut Microbiome and Cognitive Function

The term “gut microbiome” refers to the catalog of microbes and their genes that dwell in the gut [150]. These microbes, also called “microbiota,” are the microbial taxa that make up the gut microbiome [150]. Whereas “microbiome” refers to the collection of genes of the microbiota, “microbiota” refers to particular species of bacteria. Determining the health of the gut microbiome is difficult considering it contains 10–100 trillion microbial cells, although some patterns have emerged with the use of DNA sequencing [151]. The gut microbiome is considered healthy when it benefits human digestive functioning and is operating in symbiosis; this occurs when there is a higher proportion of beneficial microbiota (i.e., Firmicutes) than detrimental microbiota (i.e., Bacteroidetes) with the ideal ratio being 10:1 [152]. Alternatively, the gut microbiome is considered unhealthy (i.e., in dysbiosis) when it comprises an atypical ratio of microbiota, specifically, an increase in Bacteroidetes and a decrease in Firmicutes [153]. Dysbiosis of the gut microbiome has been associated with various pathologies known to impact cognitive function, including anxiety and depressive symptoms in animal models [154,155], MCI, and dementia [156,157], and it becomes more common with advancing age [158,159]. Like PA, the gut microbiome appears to impact cognitive function through direct and indirect pathways.

### 3.1. Direct Pathways between the Gut Microbiome and the Brain

The gut microbiome impacts cognitive function through the gut-brain-microbiome axis. The gut-brain-microbiome axis is an interconnected system of communication and processes involving the gut, the brain, and the gut microbiome. Within this system, the gut microbiome engages in bidirectional signaling with the brain via manipulation of bile acids (BAs) [160] and metabolites [161].

The gut microbiome signals the brain through manipulation of BAs. BAs are produced by the gut and metabolized by microbiota, which then alter metabolic receptors and signaling patterns [162]. As BAs can change metabolic, inflammatory, and immune responses important for neuronal processes [163], their manipulation has important effects on brain health. For one, BAs were found to play a key role in the formation of amyloid-beta (Aβ) and brain volume [162]. BAs are also useful biomarkers for predicting neurodegeneration. One type of BA, glycoursodeoxycholic acid (GUDCA), was found to be 90% accurate in predicting development of AD 2–3 years later [164].

The gut microbiome also impacts the brain through manipulation of metabolites. Metabolites are small molecules within cells that are produced when gut microbiota ferment dietary fiber [165]. Short-chain fatty acid (SCFA) metabolites impact neurophysiological processes and behavior. Indirectly, SCFA metabolites induce nerve activation in the brain, affecting glucose regulation and weight control [166]. Directly, SCFA metabolites modulate neurotransmission and increase expression of enzymes, altering production of noradrenaline and dopamine [161]. Possibly through these processes, SCFA metabolites have been associated with changes in cognition and behavior. For example, injecting mice with sodium butyrate (SB; i.e., one type of SCFA) increases effort on the tail suspension task, indicating less behavioral hopelessness [167]. Additionally, injecting mice with propionic acid (i.e., another type of SCFA) increases hyperactivity and repetitive behaviors similar to those found in autism spectrum disorder (ASD) [168].

### 3.2. Indirect Influences of the Gut Microbiome on Brain Health

The gut microbiome may also influence brain health through its effects on diseases known to impact cognitive function including obesity, inflammation, and metabolic disorders.

#### 3.2.1. Gut Microbiome and Obesity

The gut microbiome has been linked to obesity. Persons with obesity were found to have worse gut microbiome composition than lean persons, including less gut microbial diversity [137] and higher proportions of harmful microbiota [169]. Furthermore, altering the gut microbiome impacts obesity. In mice, transferring gut microbial communities from obese to lean mice increased the amount of energy harvesting and adiposity in the lean mice [170]. In middle-aged women, consumption of *Rehmannia glutinosa* root (i.e., a natural probiotic that improves the gut microbiome) increased proportions of Actinobacteria and *Bifidobacterium* in the gut and led to reductions in waist circumference [171].

#### 3.2.2. Gut Microbiome and Inflammation

The gut microbiome also influences inflammatory responses. Worse gut microbiome composition (i.e., lower ratio of good to bad microbiota) has been linked with incidence of several gastroinflammatory diseases, including Crohn’s disease [172], irritable bowel disease, and celiac disease [173]. Persons with celiac disease were found to have higher proportions of harmful bacteria [174] and less diversity of beneficial microbiota (i.e., *Bifidobacterium*) [175]. These associations may be due to the impact of the gut microbiome on inflammatory processes. The gut microbiome activates immune responses associated with systemic inflammation [176]. When in dysbiosis, the gut microbiome can lead to “leaky gut,” a disorder which is so named due to increased permeability of the gut lining that allows harmful bacteria to more easily “leak” out of the gut into the bloodstream and trigger inflammation [177].

#### 3.2.3. Gut Microbiome and Metabolic Disorders

The gut microbiome is also associated with metabolic disorders. Compared with healthy children, initial onset of type 1 diabetes has been associated with notable differences in gut microbiome composition, including lower ratios of Firmicutes to Bacteroidetes and increased proportions of other harmful bacteria [178]. Gut microbiome composition in toddlers with increased risk of developing type 1 diabetes is less diverse and more unstable than that found in healthy peers [179]. Persons with type 2 diabetes show increased likelihood for gut dysbiosis and greater proportions of pathogens [180]. Improving gut microbiome composition with probiotic supplementation was found to reduce biomarkers associated with metabolic disorders (i.e., fasting glucose, fasting plasma insulin, Homeostatic Model Assessment of Insulin Resistance (HOMA-IR)) [181], and use of prebiotic supplementation has been associated with increased biomarkers indicative of improved metabolic functioning (i.e., C-peptide) [182].

Despite being in the early stages of research, there is much initial evidence to suggest that the gut microbiome directly and indirectly impacts brain health through gut-brain-microbiome axis signaling and modulation of disease-related processes known to influence cognitive functioning. Given these associations, perhaps modification of the gut microbiome could be used to improve cognitive function.

#### 3.2.4. Altering the Gut Microbiome

Despite evidence for some long-term stability in the composition of the gut microbiome accounted for by race and geographical location [183,184,185], it is also changeable. For one, the gut microbiome changes during disease states. Persons with atherosclerotic cardiovascular disease show an increased abundance of harmful bacteria in the gut microbiome (i.e., Enterobacteriaceae and *Streptococcus spp*.) [186], and individuals with *Clostridium difficile* infection or irritable bowel disease often show gut dysbiosis and reduced microbial diversity [187].

The nutraceutical sodium butyrate also benefits the gut microbiome. Sodium butyrate was found to increase butyrate availability and promote the growth of butyrate-producing bacteria over time [188]. In mice, sodium butyrate was also found to improve gut microbiome imbalance, reduce levels of endotoxins, [189], and improve insulin resistance and glucose intolerance [190]. The benefits of sodium butyrate are further mediated by an increase in acetyl-CoA and optimized mitochondrial metabolism [188], both of which aid metabolic processes [191].

Diet also notably alters gut microbiome composition [192]. A poorer diet with high amounts of fat was found to increase proportions of harmful microbiota (e.g., *Bacteroides*) [193] and reduce amounts of microbiota overall [194]. Healthy diets (i.e., those containing lower amounts of saturated fat and processed sugars and higher amounts of fiber, protein, and natural sugars, such as the Mediterranean diet), however, have been found to improve the health of the gut microbiome. Higher amounts of protein intake were shown to increase microbial diversity, even over short-term intervals [195,196]. Consumption of natural sugars, such as those found in dates and cow’s milk, were found to increase proportions of beneficial microbiota [197] and decrease proportions of harmful microbiota [198]. Fiber also appears to benefit the gut microbiome, as consumption of whole-grain wheat cereal was shown to increase good microbiota more so than wheat bran cereal [199] or other nongrain cereals [200].

Prebiotic and probiotic supplementation also has direct effects on gut microbiome composition. Prebiotics are nondigestible ingredients in foods that promote growth and stimulate activities of bacterial species already residing in the gut [201]. Some common foods naturally containing prebiotics include bananas, leeks, onions, and oats [202]. Consumption of prebiotics has been associated with growth of beneficial microbiota including *Lactobacillus*, *Bifidobacterium* [203], and lactic acid bacteria (e.g., *Weissella cibaria*, *Lactobacillus brevis, Leuconostic mesenteroides*) [204]. Prebiotics were also found to increase *Bifidobacterium*, decrease *Clostridium* groups, and prevent deterioration of gut lining in HIV-1-infected adults [205]. Furthermore, in obese women, prebiotics appear to increase *Bifidobacterium*, promote fat metabolism, and reduce fat mass [206].

Probiotics also improve gut microbiome composition. Whereas prebiotics are ingredients broken down by microbiota, probiotics contain the living microbiota which confer health benefits to a host [207]. Probiotics are packaged and processed in a manner to ensure survival during the digestion process and prevent recolonization before arrival to the gut [165]. Probiotic supplementation was found to increase microbial species known to benefit host health (i.e., *Lactobacillus*, *Bifidobacterium*) and overall microbial diversity more so than other common supplements like omega-3 fatty acid [208]. Probiotics have also been used to reduce proportions of harmful microbiota [209], including pathogenic *Escherichia coli* [210], and to improve the balance of good to bad bacteria in the gut [211].

In sum, the gut microbiome can be altered through several means including disease, diet, prebiotics, and probiotics. PA also appears to modify the composition of the gut microbiome.

## 4. PA and the Gut Microbiome

### 4.1. Associations between PA and Gut Microbiome Composition

Cross-sectional research has identified associations between gut microbiome composition and physical fitness. In healthy young adults, those with better physical fitness [i.e., maximal oxygen uptake (VO_2_)] show greater taxonomic richness [212] and increased ratios of beneficial microbiota (i.e., Firmicutes) to harmful microbiota (i.e., Bacteroidetes) [213]. Conversely, greater proportions of detrimental microbiota were identified in young to middle-aged individuals with poorer physical fitness [214].

Intervention research has also begun showing that PA can modify the gut microbiome. In mice, increased amount of time spent swimming led to reduced intestinal barrier dysfunction [215]. Furthermore, in mice exposed to a 12-week intervention of low- or high-fat diet with SB or PA, gut microbiome composition was found to significantly improve with PA, irrespective of diet type [216]. PA was also shown to improve ratios of gut microbiota and prevent weight gain in mice consuming a high-fat diet [217]. Similar results have been identified in humans. One such study examined the effects of 6 weeks of exercise followed by 6 weeks of no exercise on the gut microbiome in sedentary, young adult women. It was discovered that significant changes in the gut microbiome occurred for lean women during the period of exercise, although these microbial changes reverted to baseline levels after exercise was discontinued [218].

There appears to be a positive association between gut microbiome composition and PA, such that more PA/better physical fitness is associated with, and can improve, the gut microbiome. Although this relationship has been largely studied in animal models, initial studies with humans have begun replicating these promising findings.

### 4.2. Potential Mechanisms for the Impact of PA on the Gut Microbiome

PA may improve the gut microbiome through modification of BAs, metabolites, gut permeability, and interactions with butyrate and mitochondria [188,219,220], each of which has bidirectional effects with the gut microbiome.

PA may alter the gut microbiome through manipulation of BAs. As described earlier in the text, the gut microbiome metabolizes BAs, which alters their patterns and signaling with the brain [162]. BAs can also alter the gut microbiome. In mice, it was discovered that changes in levels of BAs lead to notable modifications to the gut microbiome [221]. In persons with cirrhosis, changes in abundance and taxa of the gut microbiome are associated with changes in fecal BAs [222]. PA appears to modify levels of BAs through its impact on cholesterol metabolism. In male mice, those exposed to a voluntary running wheel for 12 weeks showed more cholesterol turnover and increased amounts of BAs compared to sedentary mice [223]. In healthy men, various types of exercise were found to effect BAs, with resistance exercise decreasing BAs and endurance exercise changing BA composition [224].

PA may also impact the gut microbiome through alteration of metabolites. As previously described, SCFA metabolites are produced by the gut microbiome during fermentation of dietary fibers [165]. Conversely, metabolites act as messengers to the immune system by carrying information about gut microbial patterns. These signals induce immune responses to initiate antimicrobial action or increased microbial tolerance [225]. These immune responses then work to change gut microbiome composition to confer greatest benefit for the host [225,226]. PA was found to notably alter the amounts of metabolites produced, with more time in and higher-intensity exercise associated with the greatest increases in metabolites [227].

PA may also change the gut microbiome by improving gut permeability. The gut microbiome plays a critical role in maintaining the integrity of the gut lining, ensuring harmful bacteria do not transmit to the rest of the body [177,205]. Conversely, a strong gut lining also protects the gut microbiome [228]. Two biomarkers indicative of healthy gut barriers are heat-shock proteins (HSPs) and lipopolysaccharides. HSPs are protective proteins present and inducible across all types of cells and species [229] that play an important role in protecting the health of the gut lining [230]. HSP levels increase after bouts of intense exercise, and they were found in greater amounts in persons who engage in consistent exercise, even when at rest [231]. As another marker of gut barrier health, greater amounts of LPS outside of the gut indicate increased permeability of the gut lining [232]. Compared with sedentary individuals, athletes show lower amounts of circulating LPS at rest, indicating reduced gut permeability [232].

PA also appears to improve gut microbiome composition and other aspects of health through multiple interactions with butyrate. As previously discussed, butyrate was found to promote the growth of butyrate-producing microbiota and other beneficial microbiota, and it was found to reduce endotoxins [188,189,190]. In turn, butyrate-producing microbiota increase levels of butyrate. PA was found to enhance butyrate-producing fecal bacteria and increase butyrate production [233]. Given the large body of data showing that the benefits of PA are mediated via improvements in mitochondrial function, some of the benefits of exercise on aging-associated processes are due to optimized mitochondrial function, including in immune cells. In turn, these effects may be mediated by an increase in butyrate [234]. In addition to its impact on gut microbiome composition, butyrate is an important immune and mitochondria regulator, with effects on immune cells at least partly mediated by optimized mitochondrial function. Butyrate also promotes the upregulation of the melatonergic pathway [235]. The promotion of melatonin potentiates the effects of PA and butyrate induction via its antioxidant and anti-inflammatory effects, as well as via its optimization of mitochondrial function and immune cell function [236]. Many athletes use melatonin to optimize the effects of exercise, while melatonin is dramatically decreased in the elderly. This gives ready links to immune senescence that has classically been associated with aging and dementia [237], as well as the many emerging medical conditions in the elderly [238].

As described above, PA also impacts the gut microbiome through its effects on mitochondria and immune cell function [239]. The gut microbiome was found to impact mitochondrial function leading to oxidation reduction (redox) balance, regulation of energy production, and regulation of immune reactions [240,241]. When specific functions of the gut microbiome are altered (e.g., fatty-acid metabolism, lipid biosynthesis), the gut microbiome also influences systematic immune functions that help control infection [242]. Conversely, mitochondria and immune cells also impact the gut microbiome. Mitochondria were shown to detect infectious microorganisms and subsequently activate immune responses to target them [243]. Mitochondria were also found to play a role in gut functioning [244] and to help protect the intestinal gut barrier [245], promoting gut microbiome health. Furthermore, mitochondrial genetic variants have been associated with specific gut microbiome compositions [241]. Similarly, alterations in immune system functioning (e.g., production of antimicrobial peptides, presence of natural killer (NK) T cells) have been associated with gut dysbiosis [246,247]. During endurance exercise, transcription factors (i.e., proteins that control the rates of transcription of genetic information from DNA to RNA) [248] alter mitochondrial function including mitochondrial DNA copy numbers, mitochondrial electron transport [249], and mitochondrial biogenesis [250]. Short bouts of intense exercise were also been to enhance recirculation of immune cells (e.g., NK cells, neutrophils) leading to improved immune defense [251]. As PA has been shown to alter mitochondrial and immune cell function, and interplays between mitochondria, immune cells, and the gut microbiome have been repeatedly identified, this may be another means through which PA impacts the gut microbiome.

Initial evidence suggests several mechanisms through which PA may improve the gut microbiome, including alteration of BAs, metabolites, improving gut permeability, and interactions with butyrate and mitochondria. Through PA and other means, it appears that improving the gut microbiome may impact cognitive function.

## 5. The Gut Microbiome and Cognitive Function

Like PA, the gut microbiome appears to have independent effects on cognitive function. Improvements to the gut microbiome, irrespective of PA, have been associated with better markers of brain health and cognitive outcomes.

### 5.1. Gut Microbiome Improves Cognitive Function in Animal Models

In mouse models, prebiotic and probiotic supplementation has been found to modify the gut microbiome and subsequently improve brain health and cognitive outcomes. Rats that are given prebiotics after surgery showed greater gut microbiome beta diversity, reduced inflammatory markers, and better performance on objective memory tasks compared with controls [252]. These findings suggest the inflammatory effects thought to induce postoperative cognitive dysfunction (POCD) may be ameliorated by improving the gut microbiome [252]. Probiotics appear to confer similar benefits. AD mice that consumed probiotics showed reduced hippocampal inflammatory expression and immune-reactive genes which are typically induced by Aβ, as well as improved performance on tasks of learning and memory [253]. Furthermore, prebiotics, probiotics, and synbiotics (i.e., combination of prebiotics and probiotics) were all shown to improve gut dysbiosis, physiological processes associated with cognitive dysfunction (i.e., systemic inflammation, insulin sensitivity), hippocampal plasticity, and cognitive performance in high-fat diet-fed rats [254].

### 5.2. Gut Microbiome Improves Cognitive Function in Studies with Humans

Improvement of the gut microbiome also appears to benefit brain health and cognitive function in humans. In persons with MCI, the Mediterranean diet (i.e., diet including high amounts of fruits, vegetables, grains, and healthy fats, commonly recommended to protect brain health) [255] was found to improve gut microbiome composition by reducing proportions of microbiota associated with the development of Aβ [256]. In persons with cirrhosis and cognitive dysfunction (Román et al., 2019) or fibromyalgia [257], consumption of probiotics led to significantly improved performance on cognitive tasks compared with controls. Furthermore, probiotics were found to alleviate psychological distress in healthy adults [258], as well as improve anxiety symptoms and increase the number and amount of strains/species belonging to beneficial microbiota (i.e., *Lactobacillus*, *Bifidobacteria*) in persons with chronic fatigue syndrome [259].

As alteration of the gut microbiome via methods other than PA (i.e., diet, prebiotics, probiotics) was found to benefit cognitive function, this suggests that the gut microbiome independently impacts the brain.

### 5.3. Gut Microbiome Plus Exercise for Cognitive Function

The independent effects of the gut microbiome on cognitive function are also supported by research showing that improving gut microbiome composition plus exercise leads to greater cognitive improvement than exercise alone. In transgenic AD mice engaged in 20 weeks of exercise, probiotic consumption, or both, mice who exercised or consumed probiotics showed decreased neuroanatomical markers of AD (i.e., number of beta amyloid plaques in the hippocampus), while only mice in the exercise plus probiotic group showed better cognitive performance on a visuospatial memory task as well [260]. Similar results have been identified in humans. In older adults who completed 12 weeks of resistance exercise training plus consumption of probiotic or placebo, persons in the exercise plus probiotic group showed significantly improved performance on a test of inhibition, reduced depressive and anxiety symptoms, and decreased BMI compared with those in the exercise plus placebo group [261]. Combining PA with additional means to improve gut microbiome composition appears to benefit cognitive function more than exercise alone.

Provided these results, as well as the previously discussed effects of PA on the gut microbiome, it appears likely that the effects of PA on cognitive function are at least partially explained by changes to the gut microbiome.

## 6. Previous Research on Mediation of the Effects of PA on Cognitive Function by the Gut Microbiome

### 6.1. Theoretical Reviews

Some researchers have posited that the gut microbiome may mediate the effects of PA on cognitive function, but no study has examined it appropriately or proposed a complete model. In a brief report, Yuan et al., 2015, reviewed evidence supporting the influence of the gut microbiome on emotion and cognition, as well as the bidirectional associations between the gut microbiome and physical fitness. Although the report suggested that the influence of exercise on the gut microbiome may lead to changes in cognitive function, no conceptual model was offered [262]. A recent systematic review by Schlegel et al., 2019, provided comprehensive evidence for associations among exercise, the gut microbiome, and cognitive function, as well as a description of proposed mechanisms underlying these interactions. The review concluded that altering the gut microbiome using exercise interventions could improve cognitive function in persons with AD and offered recommendations for future research [263]. However, the review was only theoretical in nature and did not propose a conceptual model or study outline.

### 6.2. Previous Interventional Research

Two previous studies examining the independent and combined roles of manipulation of the gut microbiome and PA on cognitive function were described earlier in this text [260,261]. Although these studies offered promising findings regarding the effects of the gut microbiome and PA on cognitive function and brain health, neither study contained enough information to determine whether the gut microbiome mediates the effects of PA on cognitive function.

The first study examined the effects of PA, probiotics, or both on cognitive outcomes in mice [260]. The methodology included all elements needed to examine respective pathways of the proposed mediation model (i.e., associations among PA, probiotics, gut microbiome, cognitive function). However, as mediation analyses were not implemented [260], it cannot be determined whether mediation occurred. Additionally, as the study was conducted in mice, the results cannot be fully generalized to humans.

In the second study which examined the effects of exercise with or without probiotics on cognitive function in older adults, it was discovered that persons in both groups (i.e., exercise plus probiotic, exercise plus placebo) showed improved performance on cognitive testing over time [261]. However, only those in the exercise plus probiotic group showed improved gut microbiome health (i.e., defecation frequency) as well [261]. Although conducted in older adults (an ideal sample for evaluating the proposed model), this study had several limitations regarding examination of the proposed mediation. First, the study used defecation frequency as the primary indicator of gut microbiome health, whereas DNA sequencing would have provided more information regarding changes to the composition of the gut microbiome [151]. Second, due to this limited marker of gut microbiome health, it cannot be determined whether the lack of change in defecation frequency in the exercise plus placebo group truly reflected no effect of exercise alone on the gut microbiome, or whether exercise may have impacted other aspects of gut microbiome composition that were not investigated. Lastly, by not including a true control group (i.e., no exercise plus no probiotic), it cannot be determined that exercise alone improved cognitive function at all, thereby neglecting the primary association in question (i.e., effects of PA on cognitive function).

Although the abovementioned studies examined some of the associations among PA, cognitive function, and the gut microbiome, their respective methodologies did not lead to clear conclusions about whether the gut microbiome mediates the relationship between PA and cognitive function. Furthermore, although the potential mediation was discussed in theoretical contexts, a comprehensive model integrating the complex associations among PA, the gut microbiome, and cognitive function, as well as their respective and related underlying biological mechanisms, has not yet been offered. To address this gap, a detailed theoretical model representing the potential mediation of the relationship between PA and cognitive function by the gut microbiome, as well as methodology to appropriately test the model, is described next.

## 7. Proposed Mediation Model

### 7.1. Summary of What We Know

There is ample evidence to suggest that PA impacts cognitive function. Numerous interventions have shown that exercise benefits the brain by reducing risk of/symptoms associated with various diseases (i.e., cardiovascular disease, obesity, diabetes, depression) and/or by increasing neurochemicals (i.e., BDNF, IGF-1) and related neuronal growth processes (i.e., neurogenesis, neuroplasticity, angiogenesis). Novel research proposes that the gut microbiome may impact cognitive function as well. Similar to PA, the gut microbiome has been found to benefit the brain by countering disease-related processes associated with cognitive dysfunction (i.e., obesity, inflammation, metabolic disorders) and by altering molecular activities (i.e., BAs, metabolites) that promote helpful gut-brain-microbiome axis signaling. The composition of the gut microbiome is malleable and can be improved through several different means (i.e., disease, diet, prebiotics, probiotics). There is also evidence to suggest that the gut microbiome is modifiable by PA, potentially through alteration of BAs, metabolites, and gut permeability. Improving the gut microbiome has been found to have beneficial psychological and cognitive effects.

### 7.2. Rationale

If PA and the gut microbiome have independent effects on cognitive function, and if PA changes the composition of the gut microbiome, the gut microbiome may mediate the relationship between PA and cognitive function. The gut microbiome and PA appear to share some overlapping pathways known to impact cognitive function (i.e., disease-related processes, diseases associated with cognitive dysfunction), as well as unique pathways that either do not overlap or have not yet been connected to one another (i.e., respective effects on different neurochemical processes). As the gut microbiome and PA respectively influence unique processes associated with cognitive function, and as the gut microbiome can be altered by means other than exercise, the potential mediating effect of the gut microbiome on the relationship between PA and cognitive function is likely partial, not complete.

### 7.3. Description of Proposed Mediation Model

Given the evidence reviewed thus far, a proposed model of the partial mediation of the relationship between PA and cognitive function by the gut microbiome is described below.

#### 7.3.1. PA Pathways

There are pathways currently established by the literature describing the processes through which PA improves cognitive function. These established pathways directly link PA with diseases and brain-related processes known to impact cognitive function. PA reduces the risk of developing diseases and disorders (e.g., cardiovascular disease, diabetes, obesity, depression) associated with cognitive dysfunction. PA also improves neurochemical (i.e., BDNF, IGF-1) and structural processes (i.e., neuroplasticity, neurogenesis, angiogenesis) that promote brain health and cognitive function. Through these established pathways, cognitive function appears to be protected and/or enhanced by PA.

#### 7.3.2. Gut Microbiome Pathways

Although less studied, there are also independent pathways connecting the gut microbiome with cognitive function. The composition of the gut microbiome can be altered by disease, diet, prebiotics, or probiotics. When these factors improve the gut microbiome, it subsequently reduces disease-related processes (e.g., inflammation, metabolic dysfunction, adiposity) associated with cognitive dysfunction. When the gut microbiome is in symbiosis, it also promotes healthy signaling with the brain along the gut-brain-microbiome axis by altering molecular processes (e.g., BAs, metabolites). Through these pathways, the gut microbiome appears to impact brain health and cognitive outcomes.

#### 7.3.3. Proposed Mediation Pathways

There are two possible processes through which the influence of PA on cognitive function may be mediated by the gut microbiome. The first proposed mediation pathway involves the influence of the gut microbiome on processes within the established associations among PA, disease, and cognitive function (i.e., established pathways). The second proposed mediation pathway connects PA with cognitive function through PA-induced changes to the gut microbiome (i.e., novel pathways). Each type of pathway is detailed below (See Figure A1, Appendix A).

##### Established Pathways

The gut microbiome may mediate the relationship between PA and the brain along pathways previously established in the literature. PA appears to improve cognitive function along two primary pathways: (1) reducing risk for/symptoms of diseases associated with cognitive dysfunction, and (2) improving neurochemical processing and structural changes in the brain. It is possible that PA’s effects on the gut microbiome contribute to one of these pathways. Improving the composition of the gut microbiome (using PA or other means) has been found to mitigate disease-related processes associated with cognitive dysfunction (i.e., inflammation, metabolic dysfunction, adiposity). These physiological processes are often manifestations of and/or exacerbate diseases known to impair cognitive function (i.e., obesity, diabetes, cardiovascular disease, depression). As such, by improving disease-related processes, the gut microbiome also counters the diseases themselves. If one of the primary pathways linking PA to cognitive function is through reduction of disease, and PA impacts gut microbiome health which also mitigates aspects of similar diseases, perhaps the effects of this pathway, originally thought to be a clear link between PA and cognitive function, are partially accounted for by PA’s influence on the gut microbiome. If accurate, this interpretation suggests that the respective effects of PA and the gut microbiome on disease may overlap such that the gut microbiome partially mediates the pathway connecting PA to disease and cognitive function.

##### Novel Pathways

The gut microbiome may also mediate the relationship between PA and the brain along novel pathways not previously linked to PA. It has been posited that the gut microbiome influences the brain through two primary means: (1) improving disease-related processes deleterious to brain health, and (2) altering gut-brain-microbiome axis signaling. PA improves the composition of the gut microbiome by altering BAs, gut permeability, and metabolites. In doing so, PA may spur changes in the gut microbiome that produce a cascading effect along the two primary pathways connecting the gut microbiome with cognitive function (i.e., ameliorating disease-related processes, altering gut-brain-microbiome axis signaling). As such, in addition to its independent effects on cognitive function, PA’s influence on the brain and cognitive outcomes may also be partially accounted for by its manipulation of the gut microbiome along these novel pathways.

In sum, the proposed mediation model of the effects of PA on cognitive function by the gut microbiome acknowledges the respective pathways connecting PA and the gut microbiome to the brain. Additionally, the model includes new, mediation pathways reflecting the influence of PA on the gut microbiome and its subsequent impact on the brain along previously established and/or novel pathways.

### 7.4. Testing the Proposed Mediation Model

To evaluate the accuracy of the proposed mediation model, future research should examine key features of the model and utilize statistical analyses that allow detection of partial mediation. The most appropriate study design that would enable investigation of these key features would be a randomized clinical trial.

#### 7.4.1. Key Features Needed to Examine the Proposed Mediation Model

To support the proposed mediation model, a study would need to include the following: (1) examination of all respective pathways of the model (i.e., PA to cognitive function, gut microbiome to cognitive function, PA to gut microbiome), (2) appropriate measures of key elements of the model (e.g., DNA sequencing for gut microbiome composition, neuropsychological testing for cognitive function), (3) longitudinal study design to track changes in each of the key elements over time, and (4) mediation analyses to determine which variables account for potential change in outcomes of interest (i.e., gut microbiome composition, cognitive performance). Including these requisites would allow adequate examination of the associations between changes in PA and the gut microbiome, and their effects on cognitive function. If it were discovered that changes to PA were associated with changes to the gut microbiome and cognitive function, and that changes to the gut microbiome accounted for some of the change in cognitive function associated with PA, this would support the proposed mediation model.

#### 7.4.2. Randomized Clinical Trial Description

A 2 × 2, placebo-controlled, 12-month randomized clinical trial (RCT) in older adults would be the most appropriate study design to determine whether the gut microbiome partially mediates the effects of PA on cognitive function. An RCT design reduces the risk of baseline group differences and increases the likelihood that results would be due to the intervention alone. Including a placebo reduces risk of participant expectation bias and allows clear examination of the effects of the intervention. A 2 (PA vs. no PA) × 2 (probiotic vs. placebo) design would allow examination of the multiple pathways of the proposed model, including direct associations between PA and cognitive function, between PA and the gut microbiome, between probiotics and the gut microbiome, and between the gut microbiome and cognitive function. MVPA has been consistently associated with [58,59] and found to improve [60,61,62] cognitive function. As such, MVPA would likely be the best intervention to maximize the likelihood of detecting changes to the gut microbiome and cognitive functioning. To alter the gut microbiome in a manner independent of MVPA, probiotic supplementation may be optimal. Unlike some other methods of altering the gut microbiome (e.g., changing diet), probiotic supplementation is a convenient and controlled method for altering gut microbiome composition; probiotics are easily quantifiable [264] and are simple to administer and consume [202]. To increase the likelihood that the intervention would produce detectable change in cognitive function, the trial should last 12 months (i.e., amount of time previously found appropriate for identifying cognitive change) [265,266].

As older adults have poorer gut microbiome composition compared to young adults [152], are at greater risk of developing diseases associated with cognitive dysfunction [10], and experience cognitive aging [4], they have more room for gut microbiome and cognitive improvement. Their inclusion as the primary sample would likely improve the chances of causing and detecting related changes. Given the differences in prevalence of diseases known to impact cognitive function and the gut microbiome between ethnic/racial groups (e.g., dementia, hypertension, obesity, type 2 diabetes) [97,237,267,268,269,270], as well as evidence suggesting stable differences in gut microbiome composition due to geographic location and ethnicity/race [184,185], the recruitment of a diverse sample may aid examination of the potential effects of these respective factors on outcomes. Outcomes should include measures representative of the different pathways of the proposed mediation model, including cognitive function (i.e., cognitive testing, diagnosis of MCI, AD, VaD), indicators of brain health (e.g., BDNF, brain volume), indicators of disease [e.g., BMI, glucose, tumor necrosis factor-alpha (TNF-alpha)], and gut microbiome composition (i.e., DNA sequencing for amounts/proportions of gut microbiota).

## 8. Clinical Implications of Partial Mediation by the Gut Microbiome

The possibility that the gut microbiome mediates the relationship between PA and cognitive function has important clinical implications. For instance, if the model is supported, it is likely that the effects of PA on cognitive function could be further enhanced by concurrent/adjuvant interventions to improve the gut microbiome. Both animal [260] and human trials [261] showed that probiotic supplementation and PA in tandem leads to greater improvements in cognitive function than using either intervention alone. This suggests that the cognitive benefits of PA could be augmented by concurrently improving the health of the gut microbiome using alternative methods.

The proposed mediation model also suggests that altering the gut microbiome through interventions other than PA may lead to meaningful cognitive benefits. Diet, prebiotics, and probiotics were all shown to improve gut microbiome composition, and evidence suggests that these changes additionally impact cognitive function [185,252,253,254,255,256]. As some of the cognitive benefits of PA are attributable to the gut microbiome and novel research shows the independent effects of the gut microbiome on cognitive function, altering the gut microbiome through methods other than PA (such as those described above) should also lead to cognitive gains. This finding would have considerable societal benefit, as many older adults are unable to exercise due to health or mobility issues [3,271]. Additionally, although much more research on the interactions of race/ethnicity, social determinants of health (SDOH), health outcomes, and dementia needs to be conducted and integrated into the proposed model before making informed predictions, the potential cognitive benefits gained by improving the gut microbiome without PA may feasibly extend to racial/ethnic groups who also experience limited PA due discrepancies in SDOH (e.g., less access to community amenities, less spending money for gym memberships, lower school-level SES) [272,273,274,275] and experience increased risk of dementia [267,268,276]. Thus, safe and easy methods (i.e., probiotic supplementation) that change the gut microbiome may be useful alternatives to PA that confer similar cognitive benefits.

## 9. Conclusions

This review proposed a mediation model representing what is currently known about the interactions among PA, the gut microbiome, and the brain. The proposed mediation model attempts to depict the multiple pathways connecting these factors, to what degree the pathways overlap, and the various biological mechanisms involved; it is evidently complex. Even so, the model in its present form does not yet include additional factors (e.g., race/ethnicity, SDOH, geography) that would likely interact with the proposed pathways, suggesting there is still much to be studied. Nonetheless, the possibility that the gut microbiome may mediate the relationship between PA and cognitive function offers new insights into the effects of PA on the brain and the less understood effects of the gut microbiome. Although the potential implications of this model are dependent on future research, they may inform the development of cognitive function interventions that enhance or replicate the effects of PA on the brain, particularly for older adults.

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
