# Peer review of "The Potential Mediation of the Effects of Physical Activity on Cognitive Function by the Gut Microbiome"

_geriatrics, 2020, doi:10.3390/geriatrics5040063_

Round 1

Reviewer 1 Report

Journal: Geriatrics (ISSN 2308-3417)

Manuscript ID: geriatrics-924867

Title: The potential mediation of the effects of physical activity on cognitive function by the gut microbiome

Authors: Victoria Sanborn, John Gunstad

In Section 3.2.4

It should be mentioned that the nutraceutical, sodium butyrate, is also another option for improving the gut microbiome, leading to an immediate increase in butyrate availability, as well as increasing butyrate-producing bacteria over time, with effects that are mediated via an increase in acetyl-CoA and optimized mitochondrial metabolism [Naia et al., 2017].

This would fit easily in section 4.2, and would link broader bodies of data.

Given the large body of data showing the benefits of exercise to be mediated via improvements in mitochondrial function, it should be mentioned that some of the benefits of exercise on aging-associated processes are via more optimized mitochondrial function, including in immune cells, and that this may be mediated by an increase in butyrate [Anderson and Maes, 2020]. Butyrate in particular is an important immune and mitochondria regulator, with effects on immune cells at least partly mediated via more optimized mitochondrial function. The effects of butyrate include the upregulation of the melatonergic pathway [Jin et al., 2016] and could suggest that the optimization of melatonin will potentiate the effects of physical exercise and butyrate induction, via its antioxidant and anti-inflammatory effects as well as by its optimization of mitochondrial function and immune cell function [Polyakova et al., 2018]. Most athletes use melatonin to optimize the effects of exercise, whilst melatonin is dramatically decreased in the elderly. This gives ready links to immune senescence that has classically been associated with aging and dementia [e.g. Gómez-Rubio and Trapero, in press], as well as the many emerging medical conditions in the elderly [Ciolac et al., 2020].

This may be also relevant to mention in your mediation model in Section 7, where it would easily fit.

Naia L, Cunha-Oliveira T, Rodrigues J, Rosenstock TR, Oliveira A, Ribeiro M, Carmo C, Oliveira-Sousa SI, Duarte AI, Hayden MR, Rego AC. Histone Deacetylase Inhibitors Protect Against Pyruvate Dehydrogenase Dysfunction in Huntington's Disease. J Neurosci. 2017 Mar 8;37(10):2776-2794. doi: 10.1523/JNEUROSCI.2006-14.2016.

Anderson G, Maes M. Gut Dysbiosis Dysregulates Central and Systemic Homeostasis via Suboptimal Mitochondrial Function: Assessment, Treatment and Classification Implications. Curr Top Med Chem. 2020;20(7):524-539. doi: 10.2174/1568026620666200131094445.

Jin CJ, Engstler AJ, Sellmann C, Ziegenhardt D, Landmann M, Kanuri G, Lounis H, Schröder M, Vetter W, Bergheim I. Sodium butyrate protects mice from the development of the early signs of non-alcoholic fatty liver disease: role of melatonin and lipid peroxidation. Br J Nutr. 2016 Nov 23:1-12. doi: 10.1017/S0007114516004025.

Polyakova VO, Kvetnoy IM, Anderson G, Rosati J, Mazzoccoli G, Linkova NS. Reciprocal Interactions of Mitochondria and the Neuroimmunoendocrine System in Neurodegenerative Disorders: An Important Role for Melatonin Regulation. Front Physiol. 2018 Mar 12;9:199. doi: 10.3389/fphys.2018.00199.

Gómez-Rubio P, Trapero I. The beneficial effect of physical exercise on inflammatory makers in older individuals. Endocr Metab Immune Disord Drug Targets. 2020 Jun 6. doi: 10.2174/1871530320666200606225357.

Ciolac EG, Rodrigues da Silva JM, Vieira RP. Physical Exercise as an Immunomodulator of Chronic Diseases in Aging. J Phys Act Health. 2020 May 11:1-11. doi: 10.1123/jpah.2019-0237.

Author Response

Reviewer 1 Comments:

  1. In Section 3.2.4: It should be mentioned that the nutraceutical, sodium butyrate, is also another option for improving the gut microbiome, leading to an immediate increase in butyrate availability, as well as increasing butyrate-producing bacteria over time, with effects that are mediated via an increase in acetyl-CoA and optimized mitochondrial metabolism [Naia et al., 2017].

We thank the reviewer for providing this additional information and agree that it would be a useful addition to this section of the document. We have now added a short paragraph (lines 319-324) describing the benefits of sodium butyrate for the gut microbiome and have utilized the suggested reference.

  1. This would fit easily in section 4.2 and would link broader bodies of data:

“Given the large body of data showing the benefits of exercise to be mediated via improvements in mitochondrial function, it should be mentioned that some of the benefits of exercise on aging-associated processes are via more optimized mitochondrial function, including in immune cells, and that this may be mediated by an increase in butyrate [Anderson and Maes, 2020]. Butyrate in particular is an important immune and mitochondria regulator, with effects on immune cells at least partly mediated via more optimized mitochondrial function. The effects of butyrate include the upregulation of the melatonergic pathway [Jin et al., 2016] and could suggest that the optimization of melatonin will potentiate the effects of physical exercise and butyrate induction, via its antioxidant and anti-inflammatory effects as well as by its optimization of mitochondrial function and immune cell function [Polyakova et al., 2018]. Most athletes use melatonin to optimize the effects of exercise, whilst melatonin is dramatically decreased in the elderly. This gives ready links to immune senescence that has classically been associated with aging and dementia [e.g., Gómez-Rubio and Trapero, in press], as well as the many emerging medical conditions in the elderly [Ciolac et al., 2020].”

This may be also relevant to mention in your mediation model in Section 7, where it would easily fit.

References:

Naia L, Cunha-Oliveira T, Rodrigues J, Rosenstock TR, Oliveira A, Ribeiro M, Carmo C, Oliveira-Sousa SI, Duarte AI, Hayden MR, Rego AC. Histone Deacetylase Inhibitors Protect Against Pyruvate Dehydrogenase Dysfunction in Huntington's Disease. J Neurosci. 2017 Mar 8;37(10):2776-2794. doi: 10.1523/JNEUROSCI.2006-14.2016.

Anderson G, Maes M. Gut Dysbiosis Dysregulates Central and Systemic Homeostasis via Suboptimal Mitochondrial Function: Assessment, Treatment and Classification Implications. Curr Top Med Chem. 2020;20(7):524-539. doi: 10.2174/1568026620666200131094445.

Jin CJ, Engstler AJ, Sellmann C, Ziegenhardt D, Landmann M, Kanuri G, Lounis H, Schröder M, Vetter W, Bergheim I. Sodium butyrate protects mice from the development of the early signs of non-alcoholic fatty liver disease: role of melatonin and lipid peroxidation. Br J Nutr. 2016 Nov 23:1-12. doi: 10.1017/S0007114516004025.

Polyakova VO, Kvetnoy IM, Anderson G, Rosati J, Mazzoccoli G, Linkova NS. Reciprocal Interactions of Mitochondria and the Neuroimmunoendocrine System in Neurodegenerative Disorders: An Important Role for Melatonin Regulation. Front Physiol. 2018 Mar 12;9:199. doi: 10.3389/fphys.2018.00199.

Gómez-Rubio P, Trapero I. The beneficial effect of physical exercise on inflammatory makers in older individuals. Endocr Metab Immune Disord Drug Targets. 2020 Jun 6. doi: 10.2174/1871530320666200606225357.

Ciolac EG, Rodrigues da Silva JM, Vieira RP. Physical Exercise as an Immunomodulator of Chronic Diseases in Aging. J Phys Act Health. 2020 May 11:1-11. doi: 10.1123/jpah.2019-0237.

We thank the reviewer for sharing this information about the interactions among exercise, butyrate, and mitochondrial function. We agree that adding this information to section 4.2 of the document would be beneficial for the reader and provide more comprehensive information regarding the effects of PA on the gut microbiome and other health outcomes. We now include this paragraph in the text (lines 410-426).

Reviewer 2 Report

The paper submitted by Sanborn et al. is reviewing literature on the relationship of physical activity, cognitive function and microbiome.

This paper is very interesting and attracted my attention. Nevertheless, there are some issues needed to be resolved before possible publication.

Abstract

Line 22 among instead of between

Major comments

There is lot of abbreviations used and sometimes they are not well introduced in the text where they appear for the first time. Line 110 I do not think that typical vigorous PA for the older people will be wrestling and boxing. Of note, we have to consider the osteoporosis and generally decrease of bone mass. In this light I believe this will be kind of dangerous PA for aged people. I am suggesting playing tennis etc. as a vigorous PA. In the text, all terms in latin should be in italic.

Line 234 The difference among microbiota and microbiome should be better explained. When talking about microbiota we are meaning particular species. When talking about microbiome we are considering a sum of all genes of the whole microbiota present.

Line 275 please clarify more specifically the term “worse gut microbiome” (similarly for line 283)

Line 289 please define more precisely the term “leaky gut”

Line 318 what does “healthy diet” mean? Be more specific

Line 329-330 What other kind then Lactobacillus spp. and Bifidobacteria spp. belongs to LAC (lactic acid bacteria)?

Line 338 “increase beneficial microbiota” should be better statement species exerting beneficial effect on the host health or strains/species belonging to beneficial ones

Line 340 decrease of harmful E. coli (reference 206, in this paper they are not describing pathogenic strains). Not all E. coli strains are harmful. Some strains of E. coli are commensals and some of the even probiotic ones (e.g. E. coli Nissle 1917, E. coli O83:K24:H31 and other 6 E. coli strains included in Symbioflor 2). This should be clarified and properly discussed. Probably better references should be used instead of ref. 206.

Line 427 number and amounts of strains/species belonging to beneficial microbiota instead of amounts of beneficial microbiota

Minor comments

Line 28 Prediction instead of projections

Lines 29 and 30 use plural for million (millions)

Line 54 define abbreviation VaD

Line 172 define abbreviation SES

Line 279 please indicate full name of R. glutinosa (in italic)

Line 314 clostridium – C as a capital letter and the name should be in italic

Line 406 supplementation has or supplementations have

Line 500 We letter “W” should be capital

Author Response

Reviewer 2 Comments:

The paper submitted by Sanborn et al. is reviewing literature on the relationship of physical activity, cognitive function and microbiome. This paper is very interesting and attracted my attention. Nevertheless, there are some issues needed to be resolved before possible publication.

  1. Abstract; Line 22 “among” instead of “between”

We thank the reviewer for catching this oversight. We have now updated this line to include the suggested change. (line 22)

Major Comments

  1. There is lot of abbreviations used and sometimes they are not well introduced in the text where they appear for the first time.

We apologize for this oversight. We have reviewed the text and have replaced first-time abbreviations with the full terminology throughout the document to promote clarity. 

  1. Line 110 I do not think that typical vigorous PA for the older people will be wrestling and boxing. Of note, we have to consider the osteoporosis and general decrease of bone mass. In this light I believe this will be kind of dangerous PA for aged people. I am suggesting playing tennis etc. as a vigorous PA.

We agree with the reviewer that, given the primary topics of the paper, the document should describe types of exercise that would be safe and appropriate for older adults. We have now updated this information to include vigorous exercises that would be more suitable for older adults (i.e., swimming and tennis). (line 110)

  1. In the text, all terms in Latin should be in italic.

We apologize for this oversight. We have now italicized all Latin terms in the text.

  1. Line 234 The difference among microbiota and microbiome should be better explained. When talking about microbiota we are meaning particular species. When talking about microbiome we are considering a sum of all genes of the whole microbiota present.

We have now added a line to this section further clarifying the difference between the terms “microbiome” and “microbiota.” (lines 236-237) and have reviewed the text to promote consistency in their use.

  1. Line 275 please clarify more specifically the term “worse gut microbiome” (similarly for line 283)

We apologize for the confusion. In line 275, we originally attempted to convey that “worse gut microbiome” involved characteristics described later in the same sentence, including “less gut microbial diversity and higher proportions of harmful microbiota.” To clarify, we have adjusted the wording of this sentence to more clearly convey the characteristics of “worse gut microbiome” listed above. (line 277)

In line 283, we have now specified that “worse gut microbiome” is referring to a lower ratio of good to bad microbiota. (line 286)

  1. Line 289 please define more precisely the term “leaky gut”

We thank the reviewer for this suggestion. We have now updated the line to say, “When in dysbiosis, the gut microbiome can lead to “leaky gut,” a disorder which is so named due to increased permeability of the gut lining that allows harmful bacteria to more easily “leak” out of the gut into the bloodstream and trigger inflammation” which we believe offers more clarity regarding the disorder. (lines 292-293)

  1. Line 318 what does “healthy diet” mean? Be more specific

We apologize for the lack of clarity regarding this phrasing. We have now added to this line to clarify that a healthy diet includes low levels of saturated fat and processed sugars and higher levels of fiber, protein, and natural sugars, like that found in the Mediterranean diet. (line 327-329)

  1. Line 329-330 What other kind than Lactobacillus spp. and Bifidobacteria spp. belongs to LAC (lactic acid bacteria)?

Looking into the literature more closely, there are many species of lactic acid bacteria with some of the most frequently identified being Weisella (W.) cibaria, W. confusa, Lactobacillus (Lb.) brevis, Lb. plantarum, Lb. rossiae, Leuconostoc (Leuc.) mesenteroides, Leuc. pseudomesenteroides, Lactococcus (Lc.) lactis, Enterococcus (Ec.) faecalis, and Ec. Durans (Rodríguez et al., 2019). We have now listed some of these species in the text for the reader’s convenience. (line 341)

  1. Line 338 “increase beneficial microbiota” should be better stated as species exerting beneficial effect on the host health or strains/species belonging to beneficial ones.

We apologize for the lack of clarity in the original statement. We have altered it to now say, “increase microbial species known to benefit host health.” (line 349)

  1. Line 340 decrease of harmful E. coli (reference 206, in this paper they are not describing pathogenic strains). Not all E. coli strains are harmful. Some strains of E. coli are commensals and some of the even probiotic ones (e.g. E. coli Nissle 1917, E. coli O83:K24:H31 and other 6 E. coli strains included in Symbioflor 2). This should be clarified and properly discussed. Probably better references should be used instead of ref. 206.

We thank the reviewer for catching this oversight. To promote clarity, we have adjusted the line to say, “pathogenic Escherichia coli” and have added a new reference that speaks better to the effects of probiotics on harmful E. coli. (line 352)

  1. Line 427 number and amounts of strains/species belonging to beneficial microbiota instead of amounts of beneficial microbiota.

We thank the reviewer for this recommendation. We have now updated this line to reflect the suggested terminology, namely, “increase number and amounts of strains/species belonging to beneficial microbiota.” (line 457-458)

Minor comments

  1. Line 28 Prediction instead of projections.

We have now updated this line to reflect the suggested change. (line 28)

  1. Lines 29 and 30 use plural for million (millions).

In American English, the plural “millions” is used only when nonspecific amounts are described (e.g., there are millions of people with dementia) whereas the singular “million” is used when specific amounts are described (e.g., there are 4 million people with dementia). As the manuscript in its present form is written using American English and the journal requires consistent style throughout the document, we have retained the original term “million” aligning with this language’s standards to maintain consistency.

  1. Line 54 define abbreviation VaD.

We have now added the abbreviation “VaD” to the previous line, following the full term “Vascular Dementia” to clarify the meaning of the abbreviation. (line 53) 

  1. Line 172 define abbreviation SES.

We have now added the full term “socioeconomic status” with the abbreviation “SES” following immediately after to clarify the terminology. (line 172)

  1. Line 279 please indicate full name of R. glutinosa (in italic).

We have now added the full name “Rehmannia” which is now italicized. (line 281)

  1. Line 314 clostridium – C as a capital letter and the name should be in italic

We have now capitalized and italicized “Clostridium difficile” as recommended. (line 317)

  1. Line 406 supplementation has or supplementations have.

We have now changed the word “have” to “has” to correct this error. (line 436)

  1. Line 500 We letter “W” should be capital.

We have now capitalized the word “We” as recommended. (line 530)

Round 2

Reviewer 2 Report

The authors improved the quality of the manuscript significantly after incorporation of the reviewers´ comments and suggestions. 
